# Further Characterization of the Polycrystalline p-Type β-Ga₂O₃ Films Grown through the Thermal Oxidation of GaN at 1000 to 1100 °C in a N₂O Atmosphere

Sufen Wei [1], Yi Liu [1], Qianqian Shi [1], Tinglin He [1], Feng Shi [2,*] and Ming-kwei Lee [3,*]

1 School of Ocean Information Engineering, Jimei University, Xiamen 361021, China; weisufen@jmu.edu.cn (S.W.); 202011810006@jmu.edu.cn (Y.L.); 202211810008@jmu.edu.cn (Q.S.); htl13559605621@gmail.com (T.H.)
2 Institute of Technology and Industry Research, University of Technology, Xiamen 361024, China
3 San'an Optoelectronics Co., Ltd., Xiamen 361009, China
* Correspondence: 2022000115@xmut.edu.cn (F.S.); mklee@cycu.edu.tw (M.-k.L.)

**Abstract:** The development of good-conductivity p-type β-Ga₂O₃ is crucial for the realization of its devices and applications. In this study, nitrogen-doped p-type β-Ga₂O₃ films with the characteristics of enhanced conductivity were fabricated through the thermal oxidation of GaN in a N₂O atmosphere. To obtain insights into the underlying mechanism of the thermally activated transformation process, additional measurements of the oxidized films were performed at temperatures of 1000, 1050, and 1100 °C. Room-temperature photoluminescence (PL) spectra showed a moderate ultraviolet emission peak at 246 nm, confirming the generation of gallium oxide with a band gap of approximately 5.0 eV. The characteristics of polycrystalline and anisotropic growth were confirmed via normalized X-ray diffraction (XRD), high-resolution transmission electron microscopy (HRTEM), and selected-area electron diffraction (SAED) patterns. The amount of incorporated nitrogen was analyzed via secondary ion mass spectrometry (SIMS) to examine the effects of oxidation temperature. Furthermore, the ionization energy of the acceptor in the films oxidized at 1000, 1050, and 1100 °C was calculated and analyzed using temperature-dependent Hall test results. The results indicated that nitrogen doping played a significant role in determining p-type electrical properties. The activation energy of polycrystalline β-Ga₂O₃, prepared via the thermal oxidation of GaN in the N₂O atmosphere, was estimated to be 147.175 kJ·mol⁻¹ using an Arrhenius plot. This value was significantly lower than that obtained via both the dry and wet oxidation of GaN under O₂ ambient conditions, thus confirming the higher efficiency of the thermal oxidation of GaN in a N₂O atmosphere.

**Keywords:** gallium oxide; thermal oxidation; p-type conductivity; polycrystalline crystal

## 1. Introduction

Monoclinic β-Ga₂O₃, an ultra-wide bandgap transparent semiconducting oxide, has drawn substantial attention from researchers [1–4]. Despite the notable disadvantage of its thermal conductivity, β-Ga₂O₃ applications have been observed in the fields of high-power electronics, kV class Schottky barrier diodes [5–7], e- and d-mode MOSFETs [8–10], MESFETs [11], solar-blind UV photodetectors [12,13], light-emitting diodes [14], sensing systems [15,16], solar cells [17], photocatalysts [18], and phosphors [19].

Currently, large-scale single crystals of β-Ga₂O₃ are produced using the procedures outlined by Czochralski [20] and the EFG method [21], resulting in crystals of a respectable size and structural quality. Moreover, high-quality homo-epitaxial layers have been grown using MOVPE [22] and MBE [23]. The obtained β-Ga₂O₃ crystals can exhibit behaviors characteristic of either an electrical insulator or n-type semiconductor. Via intentional doping, electron concentrations can be varied between $10^{16}$ and $10^{19}$ cm⁻³, resulting in a Hall electron conductivity ranging from $10^{-12}$ to $10^2$ S·cm⁻¹ and Hall electron mobility

of up to 170 $cm^2 \cdot V^{-1} \cdot s^{-1}$ [24–26]. However, achieving improved p-type conductivity in β-$Ga_2O_3$ has been a persistent challenge. Thus, further research on appropriate acceptor doping is necessary [27–32].

Nitrogen has emerged as a promising acceptor species for $Ga_2O_3$. Nitrogen has a slightly smaller atomic size than that of oxygen, and possesses one valence electron less than oxygen does. Previous studies have investigated the generation of p-type nitrogen-doped β-$Ga_2O_3$ films [33,34] and nanowires [35,36] through the thermal oxidation of GaN in an $O_2$ atmosphere at temperatures ranging from 1000 to 1100 °C. In our own research, we successfully grew p-type nitrogen-doped β-$Ga_2O_3$ films utilizing low-bond-energy $N_2O$ gas during the thermal oxidation of GaN [37]. Decomposing $N_2O$ into O atoms and 'N=N' effectively facilitated oxidation substitution. Consequently, the p-type β-$Ga_2O_3$ films produced in the $N_2O$ atmosphere exhibited higher Hall hole concentrations and conductivity values compared to those observed under the thermal oxidation of GaN with $O_2$ in the same temperature range.

A more detailed exploration of nitrogen doping in β-$Ga_2O_3$ is necessary. Further developing the previous study [37], this research focuses on analyzing the characteristics of β-$Ga_2O_3$ films grown at temperatures of 1000, 1050, and 1100 °C, with the aim of achieving Hall hole concentrations of above $2.55 \times 10^{16}$ $cm^{-3}$. The objectives of this study are as follows: (1) to further confirm the polycrystalline nature of the prepared films as β-$Ga_2O_3$ using PL, normalized XRD, HRTEM, and SAED techniques; (2) to investigate variations in nitrogen doping concentrations within the oxidation temperature range using secondary ion mass spectrometry (SIMS); (3) to identify the primary acceptor by analyzing the valence band spectrum and calculating the ionization energy; (4) to assess the advantages of oxidation efficiency in a $N_2O$ atmosphere by determining the activation energy, which was found to be 147.175 $kJ \cdot mol^{-1}$ in this study. For reference, the activation energies reported for the dry and wet oxidation of GaN using oxygen were 300 $kJ \cdot mol^{-1}$ [38] and 210 $kJ \cdot mol^{-1}$ [39], respectively.

## 2. Experimental Procedure

The (0001) GaN wafers used in this study were undoped and were of a native n-type character with an electron concentration of $1.02 \times 10^{17}$ $cm^{-3}$. These wafers had a thickness range of 5.8–5.9 μm. To perform the desired process, the GaN wafers were heated in a horizontal quartz tube furnace to temperatures of 1000, 1050, and 1100 °C. The process was conducted under the presence of $N_2O$ gas with a flow rate of 200 $cc \cdot min^{-1}$ for a duration of 60 min. Further information regarding the specific experimental parameters can be found in [37].

All experiments described in this research were conducted at Jiangsu 3rd Generation Semiconductor Research Institute Co., Ltd., located in Suzhou, China. Room-temperature PL spectra were measured using a fluorescence spectrometer (PI-PLE-2355/2558 + PIXIS-256E, Princeton Instruments, Trenton, NJ, USA). Two different light sources were used for the measurements, each covering a different wavelength range. For the wavelength range of 242 to 300 nm, an excitation light source of 230 nm was utilized, generated by a xenon lamp and filtered through a grating. For the wavelength range of 325 to 700 nm, a 325 nm He-Cd laser was used as the light source.

The phase structure of the samples was determined using a normalized XRD method. XRD analysis was performed using the EMPYREAN SERIES 3 instrument (Malvern Panalytical Technologies, Great Malvern, UK) with a Cu Kα (λ = 1.54 Å) X-ray source. The measurement was carried out within a 2-theta range of 10 to 80°, with a sampling pitch of 0.02° and a preset time of 0.24 s.

Transmission electron microscopy (TEM) was performed using Talos F200X S/TEM (Thermo Fisher Scientific, Waltham, MA, USA) operating at 200 kV to analyze the cross-sectional microstructure and measure oxide thickness. Cross-sectional specimens were prepared using focused ion beam (FIB) milling with a Helos-5UX instrument (Thermo

Fisher Scientific, Waltham, MA, USA). The phases and lattice structures were further examined using HRTEM and SAED.

The qualitative analysis of Ga, O, and N depth profiles in the oxide films at 1000, 1050, and 1100 °C was carried out using SIMS with a TOF.SIMS5-100 instrument (ION-TOF, Münster, Germany). During the SIMS measurements, secondary ions were detected by bombarding with 5 keV Cs+ ions at an incidence angle of 60°. To investigate the valence band characteristics of β-Ga$_2$O$_3$, X-ray photoelectron spectroscopy (XPS) was conducted using a ESCALAB Xi+ instrument (Thermo Fisher Scientific, Waltham, MA, USA). Before the XPS measurement, an applied voltage of 15 kV was used in order for the Ar ion beam used in this experiment to sputter the samples at a rate of 0.10 nm·s$^{-1}$ for 100 s, resulting in an etched depth of 10 nm. The XPS analysis provided insights into the valence band characteristics of the β-Ga$_2$O$_3$ layer. Additionally, XPS analysis confirmed that the N-doping ratio and Ga/O ratio varied with the oxidation temperature.

To establish good ohmic contact, indium electrodes were sputtered onto the films, which were then subjected to rapid thermal annealing (RTA) at 600 °C for 60 s. The electrical properties were evaluated using temperature-dependent (70 to 540 K) and magnetic-field-dependent Van der Pauw Hall effect measurements conducted using a PPMS DynaCool-9 instrument, under vacuum conditions. Based on the test results, the acceptor ionization energies were determined by calculating the slope of the linear regression formula of ln(p) versus 1000/T.

## 3. Results and Discussion

The nature of the grown films as a function of the oxidation temperature was determined through room-temperature PL spectrum measurements. Figure 1a depicts the PL spectra (at a wavelength range of 242 to 300 nm) of the samples obtained via thermal oxidation at 1000, 1050, and 1100 °C. Emission was excited at 230 nm using a xenon lamp with a grating for filtering. Due to the relatively weak intensity of the excitation light in the wavelength range of 242 to 300 nm, the sensor used to detect the photoluminescence was set to a higher amplification factor. A distinct emission peak at approximately 246 nm (~5.0 eV) was observed in all grown films. This emission peak was identified as the band-to-band luminescence of gallium oxide, confirming the successful formation of Ga$_2$O$_3$ through the thermal oxidation of GaN. Furthermore, the PL peak intensity values at 246 nm were measured for each sample, and are as follows: 7959.05 for oxidation at 1000 °C, 13,644.3 for oxidation at 1050 °C, and 21,223.48 for oxidation at 1100 °C. These values indicate that the peak intensity of PL increased by a factor of 2.6 from 1000 °C to 1100 °C. Additionally, the full width at the half-maximum of the peak significantly decreased with the increasing oxidation temperature, suggesting that the oxide layer became thicker. In Figure 1b, the PL spectra of the oxidation films are shown alongside those if an undoped commercial GaN substrate for comparison, within a wavelength range of 325 to 700 nm, using a 325 nm He-Cd laser as the light source. The emission peaks at 355 nm (~3.491 eV) and 366 nm (~3.387 eV) were attributed to the GaN material [40]. As the temperature of thermal oxidation increased, the two emission peaks of GaN decreased. Moreover, compared to the PL from the GaN substrate, the yellow emission peak at 565 nm (~2.194 eV) in the oxide films was attributed to the presence of Ga vacancies under high-temperature growth [41,42].

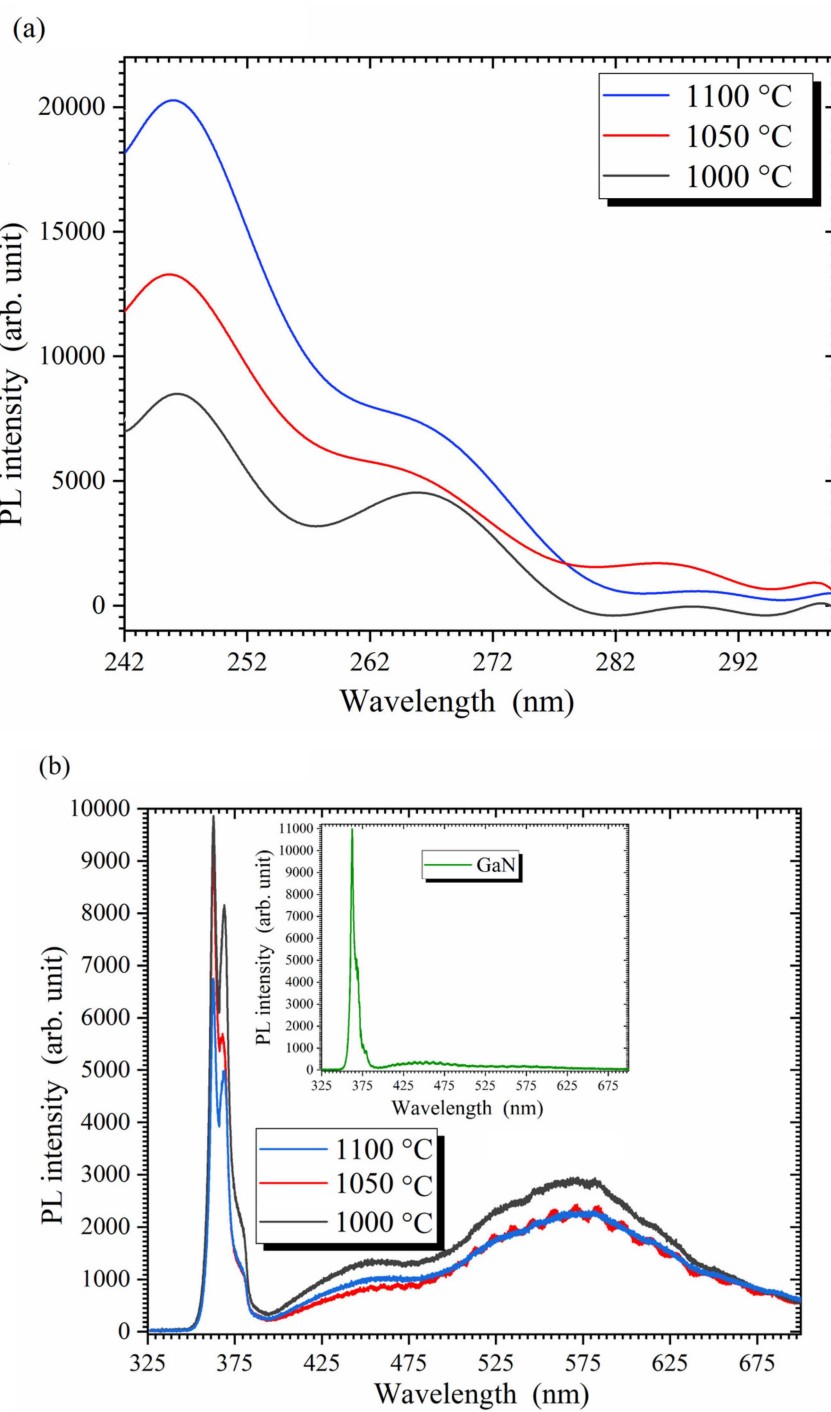

**Figure 1.** (**a**) PL emission spectra of the samples thermally oxidized at 1000, 1050, and 1100 °C, measured at room temperature, in the wavelength range of 242 to 300 nm. Excitation of the emission was performed at 230 nm using a xenon lamp with grating filtering. (**b**) PL emission spectra of the samples thermally oxidized at 1000, 1050, and 1100 °C, measured at room temperature, in the wavelength range of 325 to 700 nm. The inserted image represents the PL spectrum of the undoped GaN substrate. Excitation of the emission was performed using a 325 nm He-Cd laser.

The XRD patterns of the grown films were analyzed to determine the formation of the β-Ga$_2$O$_3$ phase and its preferred orientation along the {$\bar{2}$01} directions as a function of the oxidation temperature. The XRD results were compared with the standard powder diffraction files from COD-2004987: PDF# 43-1012 for β-Ga$_2$O$_3$, PDF# 50-0792 for GaN, and PDF# 50-1496 for sapphire. The space group of β-Ga$_2$O$_3$ is C2/m. a = 12.23 Å,

b = 3.04 Å, and c = 5.80 Å. α = γ = 90°, and β = 103.8°. In Figure 2, to account for differences in oxidation depth and the corresponding XRD intensities, each diffraction peak was normalized by the highest diffraction peak observed at the same oxidation temperature. For oxidation temperatures between 1000 and 1050 °C, the most preferred orientation was ($\bar{2}$01), located at 18.7°. At 1100 °C, the most preferred orientation was ($\bar{4}$02), located at 38.2°. In addition to the {$\bar{2}$01} family of planes at 18.7°, 38.2°, and 59.1°, there were relatively strong diffraction peaks corresponding to the (020), (002), ($\bar{1}$12), and {400} planes of β-Ga$_2$O$_3$. This suggests that the thin β-Ga$_2$O$_3$ film produced through high-temperature thermal oxidation was polycrystalline. To facilitate further analyses of the polycrystalline properties of the β-Ga$_2$O$_3$ films oxidized at 1000, 1050, and 1100 °C based on the XRD pattern, grain sizes, and the spacing (d-spacing) of the crystal plane are detailed in Tables 1 and 2. Comparing the grain sizes, it can be observed that the {$\bar{2}$01} plane had the preferred orientation at all three oxidation temperatures. For the ($\bar{4}$02) plane, oxidation at 1100 °C resulted in the largest grain size and the highest crystal quality. Moreover, in addition to the dominant {$\bar{2}$01} plane β-Ga$_2$O$_3$ orientation, other β-Ga$_2$O$_3$ peaks, such as (020), (002), ($\bar{1}$12), and {400}, became more pronounced for oxidation temperatures ≥1050 °C. These findings are consistent with the observations made through TEM and SAED. In summary, the higher the oxidation temperature, the greater the intensities of the diffraction peak.

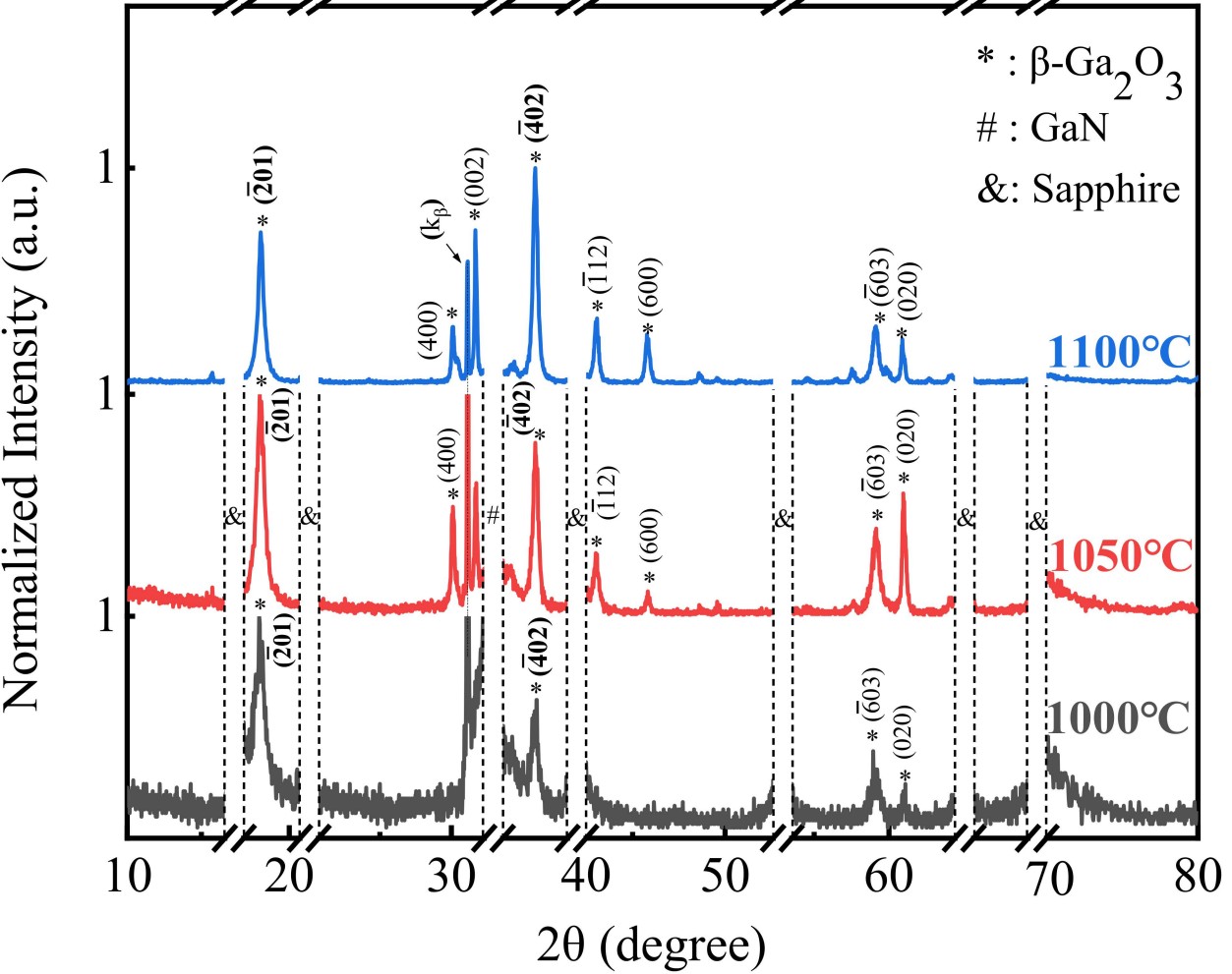

**Figure 2.** Normalized XRD patterns of samples thermally oxidized at 1000, 1050, and 1100 °C compared with ICDD file PDF# 43-1012 for β-Ga$_2$O$_3$, PDF# 50-0792 for GaN and PDF# 50-1496 for sapphire.

**Table 1.** Grain sizes and the crystal plane spacing (d-spacing) of {$\overline{2}$01} directions from the XRD pattern.

| Oxidation Temperature (°C) | ($\overline{2}$01) Grain Size (nm) | ($\overline{2}$01) d-Spacing (nm) | ($\overline{4}$02) Grain Size (nm) | ($\overline{4}$02) d-Spacing (nm) | ($\overline{6}$03) Grain Size (nm) | ($\overline{6}$03) d-Spacing (nm) |
|---|---|---|---|---|---|---|
| 1000 | 66.59 | 4.6790 | 16.78 | 2.3420 | 11.14 | 1.5597 |
| 1050 | 76.59 | 4.6790 | 31.02 | 2.3420 | 17.12 | 1.5597 |
| 1100 | 80.95 | 4.6790 | 84.97 | 2.3420 | 19.47 | 1.5597 |

**Table 2.** Grain sizes and the crystal plane spacing (d-spacing) of other directions from the XRD pattern.

| Oxidation Temperature (°C) | (020) Grain Size (nm) | (020) d-Spacing (nm) | (002) Grain Size (nm) | (002) d-Spacing (nm) | ($\overline{1}$12) Grain Size (nm) | ($\overline{1}$12) d-Spacing (nm) | (400) Grain Size (nm) | (400) d-Spacing (nm) | (600) Grain Size (nm) | (600) d-Spacing (nm) |
|---|---|---|---|---|---|---|---|---|---|---|
| 1000 | 12.15 | 1.5200 | NA | NA | NA | NA | 7.25 | 2.9710 | NA | NA |
| 1050 | 33.12 | 1.5200 | 35.91 | 2.8170 | 24.13 | 2.0980 | 31.59 | 2.9710 | 31.96 | 1.9793 |
| 1100 | 35.04 | 1.5200 | 44.54 | 2.8170 | 34.07 | 2.0980 | 33.29 | 2.9710 | 33.06 | 1.9793 |

Figure 3a–c depict the SAED pattern obtained from a detailed characterization of the grown β-Ga$_2$O$_3$ film. The discrete bright spots observed in the pattern correspond to β-Ga$_2$O$_3$ crystal planes ($\overline{2}$01), ($\overline{4}$02), (002), (111), (020), and (400). These findings provide further evidence of the polycrystalline nature of the thin β-Ga$_2$O$_3$ film. Furthermore, the HRTEM images of the region near the β-Ga$_2$O$_3$/GaN interface for the samples thermally oxidized at 1000, 1050, and 1100 °C are shown in Figure 3d–f. These images reveal a distinct boundary between the β-Ga$_2$O$_3$ film and the GaN substrate, without a transition zone. Moreover, the crystal plane orientations observed in the HRTEM images confirm the polycrystalline nature of the thin β-Ga$_2$O$_3$ film.

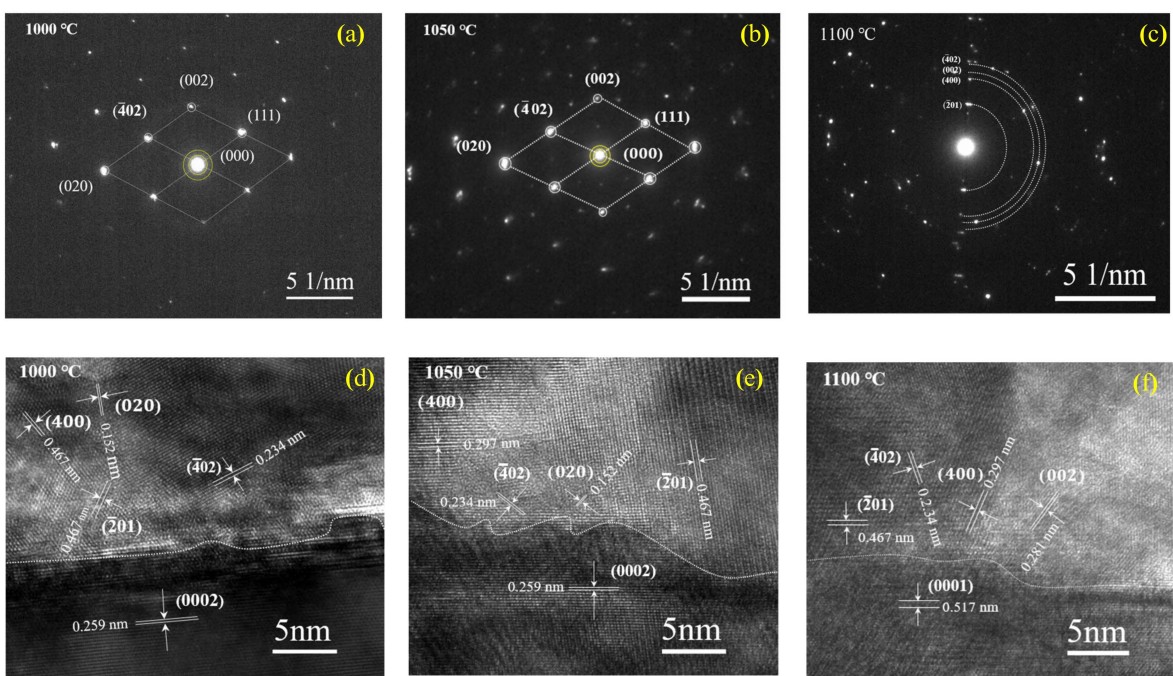

**Figure 3.** The SEAD patterns of the β-Ga$_2$O$_3$ layer grown at (**a**) 1000 °C, (**b**) 1050 °C, and (**c**) 1100 °C, and the corresponding HRTEM images taken from the β-Ga$_2$O$_3$/GaN interface of the thermally oxidized samples (**d**) 1000 °C, (**e**) 1050 °C, and (**f**) 1100 °C.

In this study, the oxidation thickness from [37] was used to determine the activation energy required for the thermal oxidation of (0001) monocrystalline GaN to form polycrystalline β-Ga$_2$O$_3$ in the N$_2$O atmosphere. The activation energy represents the minimum energy necessary for initiating a reaction process. The temperature dependence of chemical reactions is often described using the Arrhenius equation, which was utilized in this study to calculate the activation energy of GaN oxidation in a N$_2$O atmosphere. It is important to note that for the purposes of this study, it was assumed that the activation energy remained constant across the temperature range of 900 to 1100 °C. The Arrhenius equation, expressed in exponential form, is represented by Equation (1):

$$Thickness = A \times e^{(-Ea/R \times T)} \tag{1}$$

where *Thickness* represents the thickness of the oxide layer at various temperatures. *Ea* denotes the activation energy in kJ/mol. The constant *A* is referred to as the pre-exponential factor in the Arrhenius equation, representing a constant unique to each activation process, but of which the specific value is unknown in this study. *T* denotes the oxidation temperature measured in Kelvin. Additionally, *R* represents the molar gas constant, which is equal to the product of the Boltzmann constant and Avogadro's constant.

To derive the logarithmic form of the Arrhenius formula, Equation (1), we proceed by taking the natural logarithm of both sides of the equation. This yields Equation (2):

$$\ln(Thickness) = \ln(A) - \frac{Ea}{R \times T} \tag{2}$$

Figure 4 presents a cumulative plot of the natural logarithm values of the oxide layer's thickness (ln(*Thickness*)) in nm as a function of the reciprocal of the oxidation temperature ((−1000)/T) in Kelvin for samples oxidized at temperatures of 900, 950, 1000, 1050, and 1100 °C. It is evident from Figure 4 that the plotted curve does not exhibit a completely linear relationship within the oxidation temperature range of 900 to 1100 °C. However, for the purposes of this study, an idealized assumption was made, in that the activation energy in Equation (2) was assumed to remain constant regardless of the oxidation temperature. Based on this assumption, Equation (2) was used to calculate the thickness of the oxide layer at two different temperatures (*T_1* and *T_2*), resulting in the derivation of Equation (3) which eliminates the unknown constant *A*:

$$\ln(Thickness\_1) - \ln(Thickness\_2) = \frac{Ea}{R \times T\_1} - \frac{Ea}{R \times T\_2} \tag{3}$$

The values for *Thickness*_1 and *Thickness*_2 represent the oxide layer thickness of β-Ga$_2$O$_3$ at 900 °C and 1100 °C, respectively. Utilizing Equation (3), the calculated activation energy was determined as approximately 147.175 kJ·mol$^{-1}$. Previous studies featuring both the dry and wet thermal oxidation of GaN in an O$_2$ atmosphere reported activation energies of 300 kJ·mol$^{-1}$ [38] and 210 kJ·mol$^{-1}$ [39], respectively. This finding demonstrates that in a N$_2$O atmosphere, the O atom with a lower bond energy can more easily dissociate from the covalent bond at the same temperature. Consequently, the oxidation rate is relatively rapid, and the activation energy required is comparably lower.

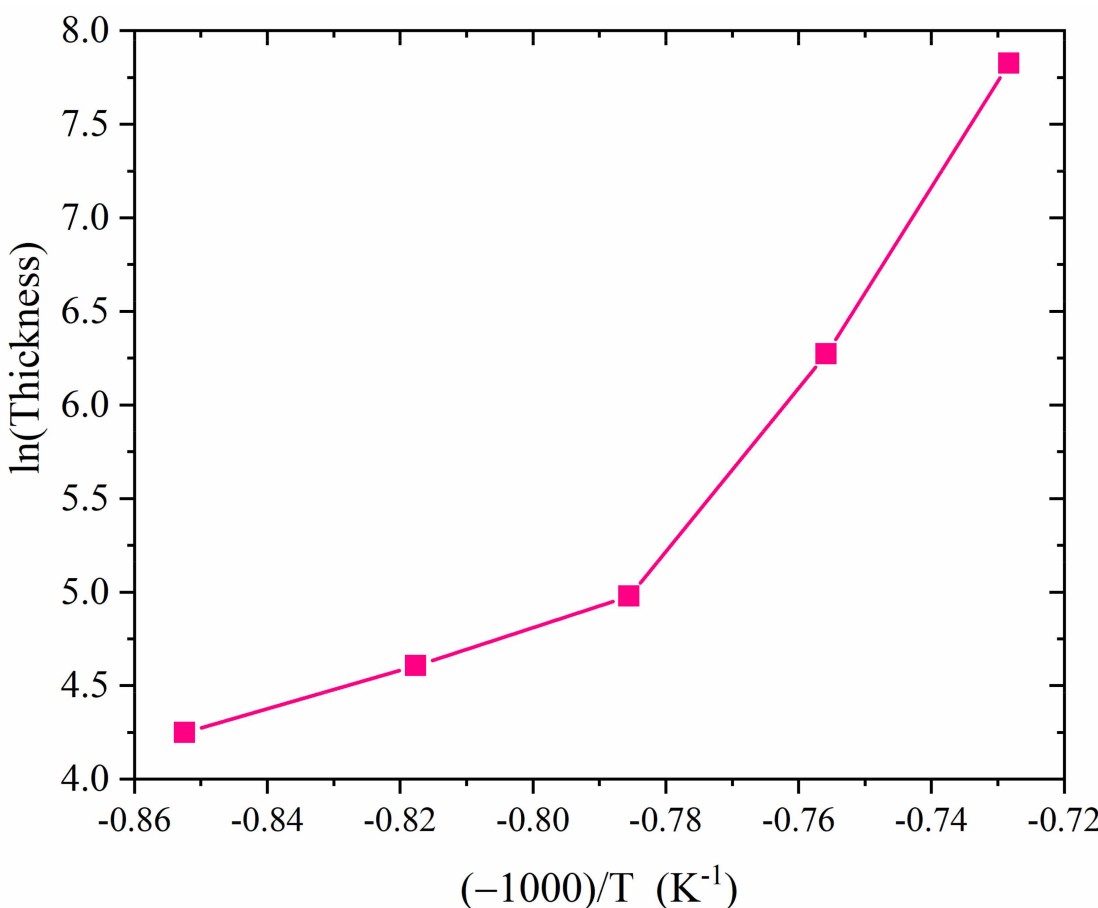

**Figure 4.** The natural logarithm plot of the oxide layer thickness (ln(*Thickness*)) in nm as the function of oxidation temperature ((−1000)/T) in Kelvin for the samples thermally oxidized at 900, 950, 1000, 1050, and 1100 °C.

Figure 5a–c display the vertical distributions of O, Ga, and N elements obtained from SIMS for samples subjected to thermal oxidation at temperatures of 1000, 1050, and 1100 °C. The depth of the analysis extends from the top surface layer to a depth of 2.5 μm. Figure 5a shows the O ion response intensities of the SIMS test as a function of depth. For a clearer display, the three temperature-oxidized samples summarized in Figure 5a are shown separately in three subfigures of Figure A1a–c in the Appendix A, which shows the O ion response intensities of the SIMS test as a function of the depth of β-$Ga_2O_3$ films at 1000, 1050, and 1100 °C, respectively. Based on the FIB-TEM measurements [37], the thickness of the β-$Ga_2O_3$ films at 1000, 1050, and 1100 °C was determined to be 145 nm, 530 nm, and 2.5 μm, respectively, which is consistent with the longitudinal distribution of O shown in Figure 5a and Figure A1a–c in the Appendix A. Consequently, it was observed that within the β-$Ga_2O_3$ layer, the O content gradually decreased towards the β-$Ga_2O_3$/GaN interface. Moreover, a higher oxidation temperature led to a greater reduction in O content near the β-$Ga_2O_3$/GaN interface. This reduction is attributed to the migration of N atoms from the underlying GaN during thermal oxidation, resulting in N-doped β-$Ga_2O_3$ and substituting O atoms. Additionally, higher temperatures promoted the release of O atoms from the covalent bond within the polycrystalline β-$Ga_2O_3$ layer, creating vacancies. Figure 5b presents the response intensities of Ga ions obtained via SIMS characterization as a function of depth. Additionally, for a clearer display, Figure A2a–c in the Appendix B showed the Ga ion response intensities determined via the SIMS test as a function of the depth of β-$Ga_2O_3$ films at 1000, 1050, and 1100 °C, respectively. It was observed that the stoichiometric ratio of Ga:O in the grown β-$Ga_2O_3$ layer was 2:3, while the stoichiometric ratio of Ga:N in the GaN substrate was 1:1. Notably, the content of Ga was relatively low

within the depth range of the corresponding β-Ga$_2$O$_3$ layer, while it was relatively high within the depth range of the corresponding GaN layer. Furthermore, the depth at which the change in Ga content occurs coincides with the observed oxidation thickness from FIB-TEM, confirming the successful oxidation of the β-Ga$_2$O$_3$ layer. It can also be observed from Figure A2a,b in the Appendix B that when the oxidation temperatures were 1000 and 1050 °C, within the range of the β-Ga$_2$O$_3$ layer, there was a "high-low-high" variation trend in the intensity of Ga as characterized via SIMS. Near the middle depth of the β-Ga$_2$O$_3$ layer, the Ga content was the lowest, indicating the presence of Ga vacancies resulting from high-temperature oxidation. Additionally, G atoms separated from β-Ga$_2$O$_3$ moved upwards to surface. In Figure A2c in the Appendix B, when the oxidation temperature was 1100 °C, the Ga content at the top of the β-Ga$_2$O$_3$ film was lowest. For the sample thermally oxidized at 1100 °C, the G atoms decomposed from β-Ga$_2$O$_3$ moved outwards. Figure 5c presents the response intensities of N ions as a function of the depth as obtained via SIMS characterization. Additionally, for a clearer display, Figure A3a–c in the Appendix C shows the N ion response intensities determined via the SIMS test as a function of the depth of the β-Ga$_2$O$_3$ films at 1000, 1050, and 1100 °C, respectively. The position where the N content stabilizes corresponds to the interface between the β-Ga$_2$O$_3$ and GaN layers. Moreover, a portion of N within the β-Ga$_2$O$_3$ region decomposes from GaN and diffuses into the overlying β-Ga$_2$O$_3$ layer, leading to an increased relative concentration of N. The N-doping concentration is the highest close to the GaN layer and decreases further away from it. Regarding N atoms separated from N$_2$O, Figure 5d provides an enlarged view of the N content at depths ranging from 0 to 20 nm for samples thermally oxidized at 1000, 1050, and 1100 °C. Although N atoms decomposed from N$_2$O contribute to N-doping in the β-Ga$_2$O$_3$ layer, their depth of penetration is shallow, and the doping concentration is low at a depth of 20 nm. The N-O bond in N$_2$O is relatively weak, facilitating the separation of O atoms at high temperatures. In contrast, the double-bonded N atoms in N$_2$O require a higher bond energy to break, resulting in a lower proportion of N being separated at high temperatures. Consequently, it is evident that the main source of N doping in the grown β-Ga$_2$O$_3$ layer is the decomposition of N from GaN.

To further validate the observed proportions of elements revealed by the SIMS tests, the elemental ratios of N and Ga/O were analyzed using XPS and are presented in Table 1. The stoichiometric ratio of Ga/O in pure β-Ga$_2$O$_3$ is approximately 0.667. However, the polycrystalline β-Ga$_2$O$_3$ grown through thermal oxidation consistently exhibited an O-deficient state, as indicated by the Ga/O ratios in Table 3. During the oxidation process, although O vacancies were formed, the main reason for the O deficiency is the replacement of O by N rather than the presence of O vacancies. Moreover, as the oxidation temperature increased, a slight decrease in the proportion of Ga suggests a corresponding increase in Ga vacancies. Furthermore, with increasing oxidation temperature, the proportion of N in the β-Ga$_2$O$_3$ layer also increased, which is in line with the results of the N response intensities observed in SIMS. This finding further confirms that the films exhibit improved p-type conductivity with higher oxidation temperatures.

**Table 3.** The elemental ratio of N and Ga:O ratio values determined via XPS.

| Oxidation Temperature (°C) | 1000 | 1050 | 1100 |
|---|---|---|---|
| Ga/O ratio | 0.831 | 0.821 | 0.818 |
| N elemental ratio (%) | 0.032 | 0.037 | 0.088 |

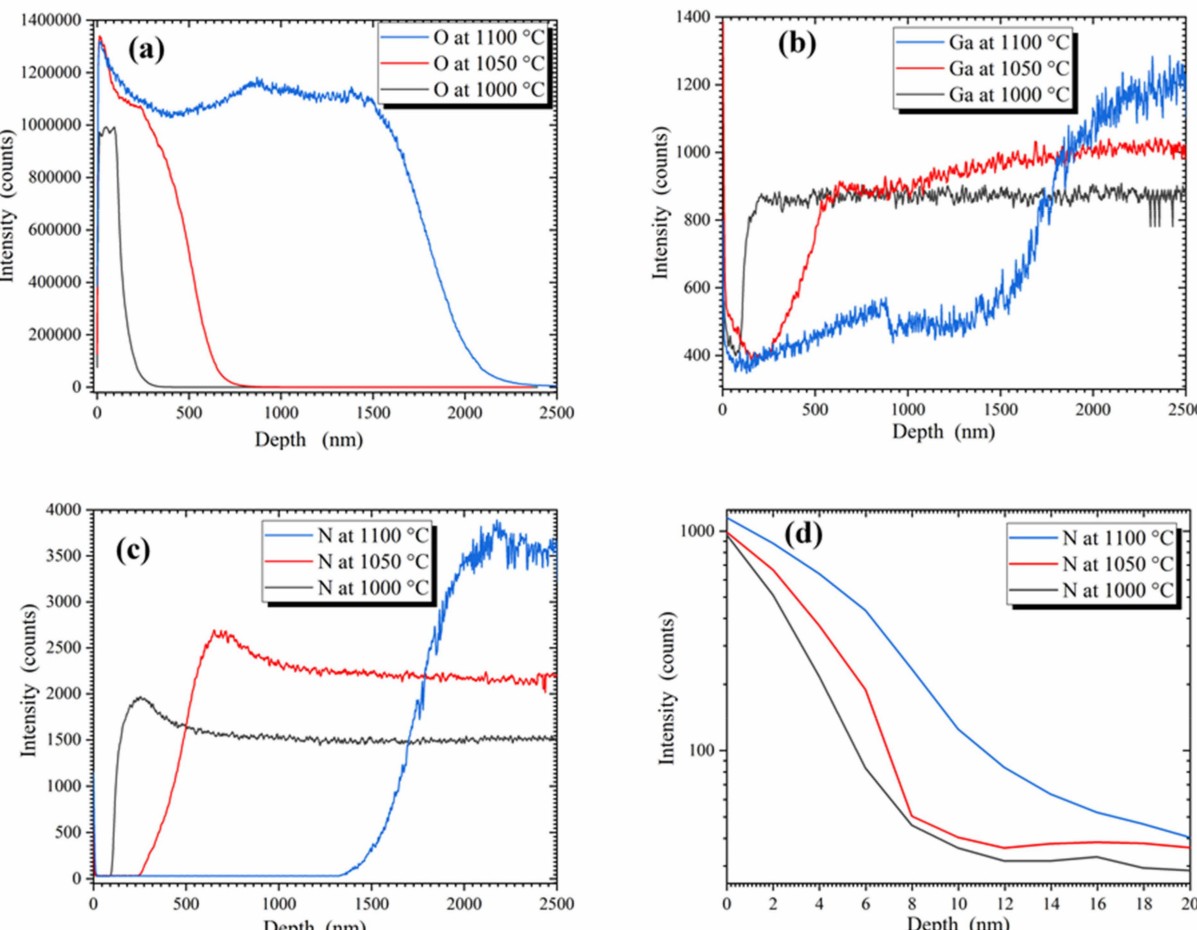

**Figure 5.** SIMS characterizations of the depth profiles of (**a**) O, (**b**) Ga, and (**c**) N from a depth of 0 to 2.5 µm for the samples thermally oxidized at 1000, 1050, and 1100 °C. (**d**) The enlarged view of N with a depth of 0–20 nm.

Figure 6a–e presents the valence band characteristics of the samples thermally oxidized at 1000, 1050, and 1100 °C, which were investigated using XPS. Previous theoretical studies [43,44] and experimental findings [45,46] have shown that the density of states in the valence band is primarily attributed to the characteristics of O 2p. The Ga 3d peak was used as a calibration peak. With an increasing oxidation temperature, the valence band edge of O 2p shifted further to the right (Figure 6a,b), indicating a lower Fermi level and a more distinct p-type nature. Moreover, Figure 6c–e provides enlarged views of the tail parts of the valence band for the samples thermally oxidized at 1000, 1050, and 1100 °C, respectively. The y axis of Figure 6c–e, representing counts per second (counts/s), is normalized to the maximum value ranging from 0 to 5 eV. The slope variation at the turning point of the tail confirms the existence of states in the lower part of the bandgap. As the oxidation temperature increases, the shallow acceptor levels introduced by N doping move closer to the valence band, indicating a higher p-type carrier density and thus an improvement in p-type conductivity.

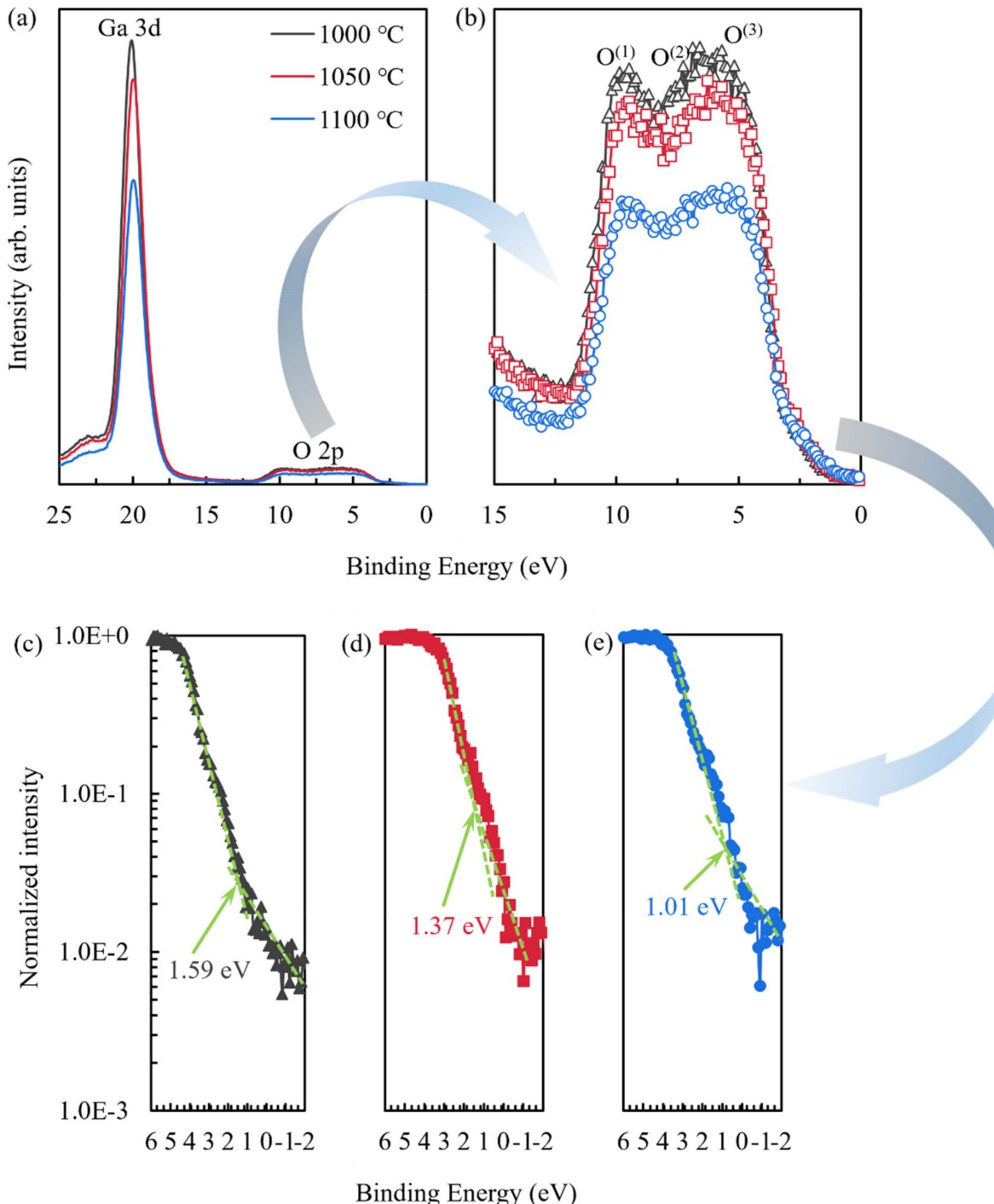

**Figure 6.** (**a**) X-ray photoelectron spectroscopy of the β-Ga$_2$O$_3$ valence band for the samples thermally oxidized at 1000, 1050, and 1100 °C (calibrated with respect to Ga 3d). (**b**) A magnified view of the valence band region for the characteristic peak O of 2p. Further investigation was performed for the tail states of the valence band for the samples thermally oxidized at (**c**) 1000 °C, (**d**) 1050 °C, and (**e**) 1100 °C.

The samples oxidized at 1000, 1050, and 1100 °C were subjected to temperature-dependent Hall measurements under vacuum conditions, ranging from 70 to 540 K. The positive Hall coefficients observed in Figure 7a confirmed the p-type nature of the N-doped β-Ga$_2$O$_3$ films. Figure 7b indicates that the f-Factor slightly increased with the test temperature, but remained close to 1, ranging from 0.98 to 0.99. Therefore, the rationality of Hall

test results was guaranteed. Figure 7c–e highlight the variations in Hall hole concentration, Hall hole mobility, and Hall resistivity, respectively, as a function of test temperature. Additionally, Figure 7f presents a natural logarithm plot of the Hall hole concentration (ln(p)) as a function of (1000/T). As the test temperature increased from 70 to 540 K, the Hall carrier concentration increased, while Hall mobility and Hall resistivity decreased. Moreover, a higher oxidation temperature resulted in greater N doping in the $\beta$-Ga$_2$O layer formed via oxidation, contributing to a higher Hall hole concentration, improved Hall mobility, and reduced Hall resistivity. At a test temperature of 300 K, the Hall hole concentrations ranged from $2.55 \times 10^{16}$ cm$^{-3}$ (for 1000 °C oxidation temperature) to $1.64 \times 10^{17}$ cm$^{-3}$ (for 1100 °C oxidation temperature). The presence of Ga and O vacancies in the $\beta$-Ga$_2$O films obtained through high-temperature oxidation was confirmed via the PL and SIMS results. These vacancies, particularly the $V_{Ga}$ and $V_{Ga}-V_O$ pairs, may act as potential acceptors [47–49]. However, based on the near-band-edge structure of $\beta$-Ga$_2$O$_3$ nanostrips depicted in Figure 7 of [50], the ionization energy of the acceptor level of $V_{Ga}$ is approximately 0.494 eV ($E_0 - E_{D1}$) [50]. At 300 K, this level of ionization energy can only contribute to hole concentration on the order of $10^{15}$. Therefore, the observed Hall hole concentration ranging from $2.55 \times 10^{16}$ to $1.64 \times 10^{17}$ cm$^{-3}$ was primarily a result of N doping in $\beta$-Ga$_2$O$_3$. The oxidation process successfully produced thin $\beta$-Ga$_2$O$_3$ films with reasonably good Hall hole concentrations, which increased proportionally with the oxidation temperature. However, although mobility also increased with higher oxidation temperatures, the improvement was not significant (Figure 2b). This phenomenon can be attributed to the generation of more voids within the cross-section of the $\beta$-Ga$_2$O$_3$ film as the oxidation temperature increased from 1000 to 1100 °C. Consequently, these voids introduced defects and dangling bonds near them, hindering the overall improvement in mobility. Therefore, no significant increase in overall mobility was observed. At a test temperature of 300 K, the Hall hole mobility was found to be 2.2 cm$^2 \cdot$V$^{-1} \cdot$s$^{-1}$ (for the 1000 °C oxidation temperature), 3.3 cm$^2 \cdot$V$^{-1} \cdot$s$^{-1}$ (for the 1050 °C oxidation temperature), and 5 cm$^2 \cdot$V$^{-1} \cdot$s$^{-1}$ (for the 1100 °C oxidation temperature). Similarly, there was a slight decrease in Hall resistivity. At a test temperature of 300 K, the Hall hole resistivity was 74 $\Omega \cdot$cm (for 1000 °C oxidation temperature), 45 $\Omega \cdot$cm (for 1050 °C oxidation temperature), and 7.7 $\Omega \cdot$cm (for 1100 °C oxidation temperature). Figure 7d presents a natural logarithm plot of the Hall hole concentration (ln(p)) as a function of (1000/T). Based on the linear regression formula of ln(p) versus 1000/T, the acceptor ionization energies of the oxidized samples at 1000, 1050, and 1100 °C were similar at approximately $0.092 \pm 0.005$ eV. The smaller acceptor's ionization energy suggests that the activation of holes occurred more easily due to N doping rather than due to Ga vacancies in the $\beta$-Ga$_2$O$_3$ film. This finding further strengthens the argument that N doping plays a dominant role in p-type electrical conductivity.

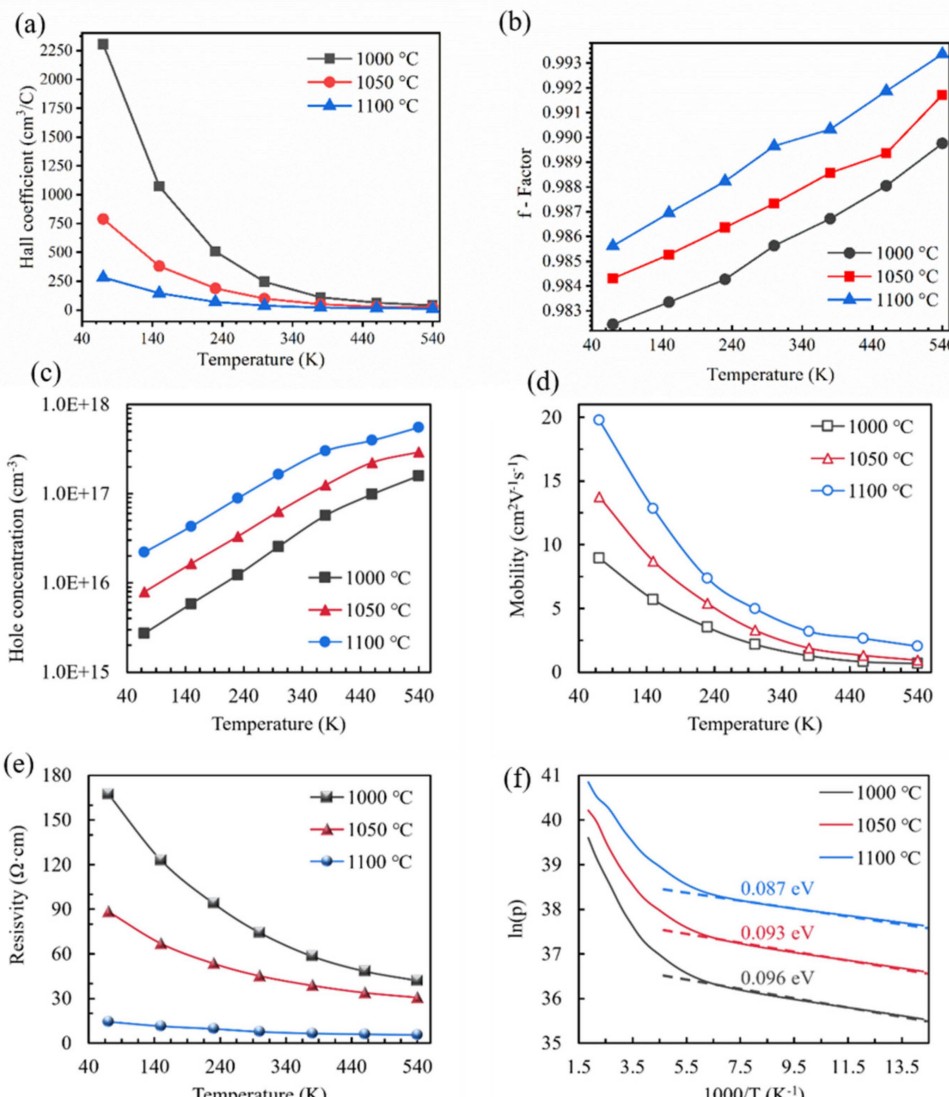

**Figure 7.** The variations in (**a**) the Hall coefficient, (**b**) f-Factor, (**c**) Hall hole concentration, (**d**) Hall hole mobility, and (**e**) Hall resistivity as a function of the test temperature for the samples thermally oxidized at 1000, 1050, and 1100 °C. (**f**) The natural logarithm plot of the Hall hole concentration (ln (p)) as the function of (1000/T).

## 4. Conclusions

Nitrogen-doped acceptors in β-Ga$_2$O$_3$ with Hall hole concentrations ranging from $2.55 \times 10^{16}$ cm$^{-3}$ to $1.64 \times 10^{17}$ cm$^{-3}$ were successfully achieved through the thermal oxidation of GaN in a N$_2$O atmosphere at temperatures ranging from 1000 to 1100 °C. Structural analysis techniques such as PL characterization, normalized XRD, and FIB-TEM confirmed that the grown β-Ga$_2$O$_3$ films were polycrystalline. Additionally, the calculation of the activation energy based on the oxidation thickness indicated that the process of thermal oxidation and nitrogen doping in the N$_2$O atmosphere was more efficient compared to the use of an oxygen atmosphere. Finally, the analysis of the valence band energy spectrum and Hall electrical properties demonstrated that the p-type conductivity of the polycrystalline β-Ga$_2$O$_3$ films was primarily achieved by substituting N for O in the β-Ga$_2$O$_3$ structure.

**Author Contributions:** S.W. designed the experimental and test schemes, organized the data, and wrote the paper. Y.L. conducted the experiments and analyzed the data and measurements. Q.S. and T.H. contributed to the data analysis and measurements. F.S. provided assistance with analyzing the fundamental principles and contributed to the writing of the paper. M.-k.L. proposed the experimental concept, guided the analysis of the experimental results, and provided input for editing the manuscript. All authors have read and agreed to the published version of the manuscript.

**Funding:** This work was supported by the Foreign Cooperation Project of Fujian Provincial Department of Science and Technology (grant no. 2020I0022).

**Institutional Review Board Statement:** Not applicable.

**Informed Consent Statement:** Not applicable.

**Data Availability Statement:** Not applicable.

**Conflicts of Interest:** The authors declare no conflict of interest.

## Appendix A

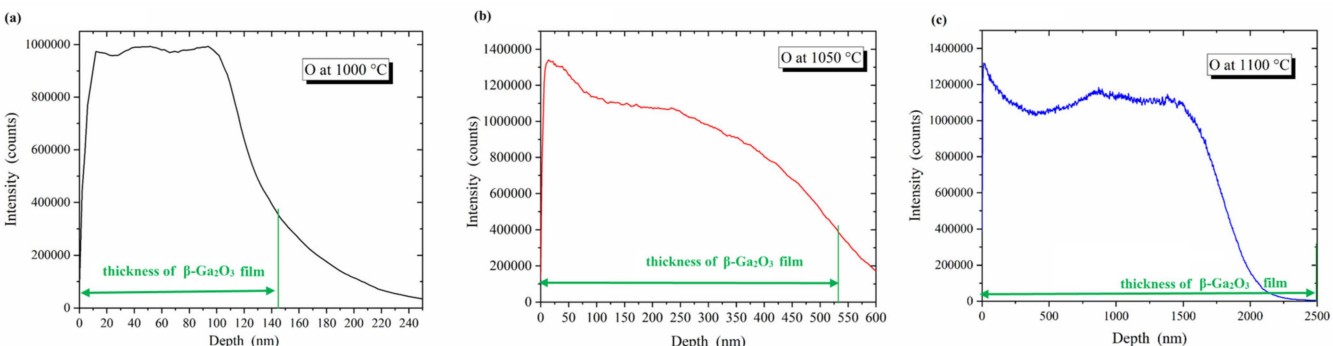

**Figure A1.** SIMS characterizations of the depth profiles of O for the thermally oxidized samples at (**a**) 1000 °C, (**b**) 1050 °C, and (**c**) 1100 °C.

## Appendix B

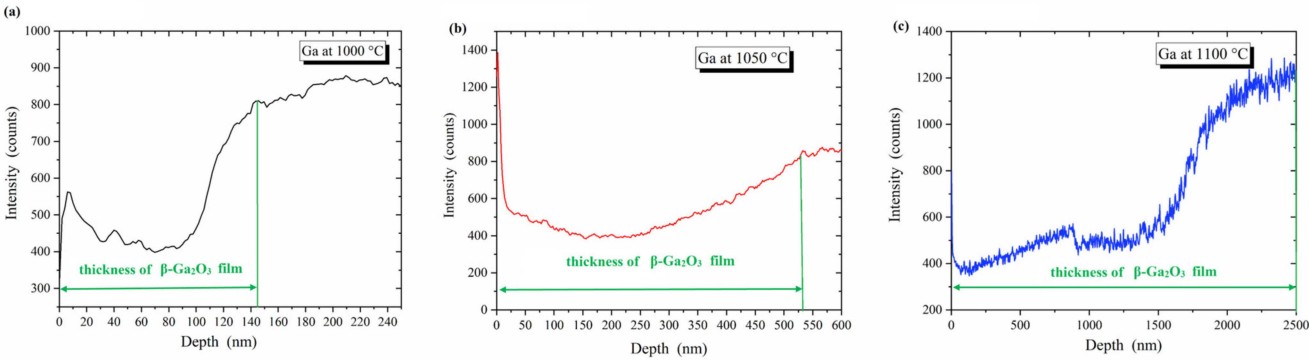

**Figure A2.** SIMS characterizations of the depth profiles of Ga for the thermally oxidized samples at (**a**) 1000 °C, (**b**) 1050 °C, and (**c**) 1100 °C.

## Appendix C

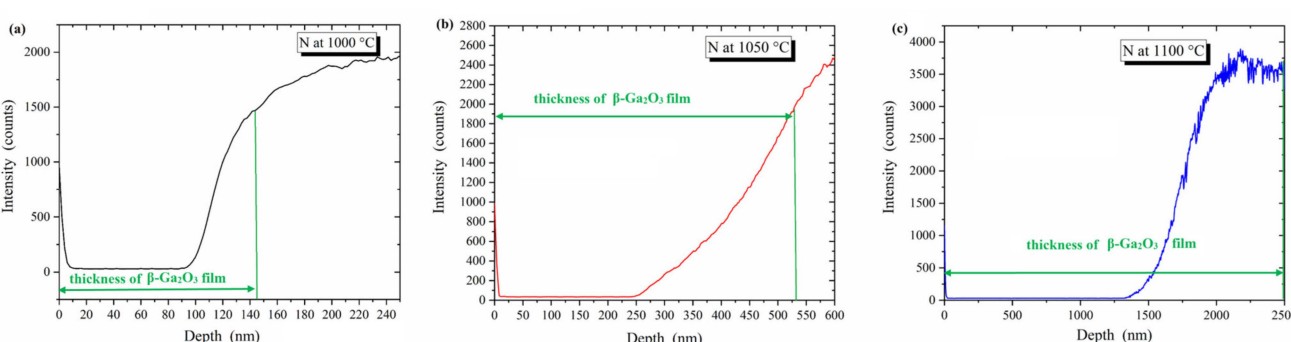

**Figure A3.** SIMS characterizations of the depth profiles of N for the thermally oxidized samples at (**a**) 1000 °C, (**b**) 1050 °C, and (**c**) 1100 °C.

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
