# Peer review of "Further Characterization of the Polycrystalline p-Type β-Ga2O3 Films Grown through the Thermal Oxidation of GaN at 1000 to 1100 °C in a N2O Atmosphere"

_coatings, doi:10.3390/coatings13091509_

Round 1

Reviewer 1 Report

In this article, a very detailed study of p-type  beta-Ga2O3 films obtained via thermal oxidation of GaN in N2O atmosphere was carried out. To become suitable for publication, the article needs additional improvement.

1. All the arguments of the authors are based on the assumption that the cause of p-type conductivity is the substitution of O-atoms by N-atoms. To confirm this, it is nececssary to additionally draw a depth distribution for the ratios Ga/O, Ga/N and O/N. This will make the reasoning presented in lines 187-223 more understandable.

2. Figure 4 needs to be corrected. The ordinate axis should be done in meters, but on a decimal logarithmic scale. This will allow the reader to better navigate the presented results. The abscissa axis indicates that the temperature is given in K, but the values clearly correspond to 1000/T(C). 

3. From what data was Figure 4 obtained? From the profiles shown in Figure 5, it follows that the thickness of the oxide layer increases with an increase in the oxidation temperature. In Figure 4, on the contrary, the thickness of the film increases with an increase of 1000/T, that is, with a decrease in temperature.

4. The numbers of the figures are mixed up in lines 181-223 (Figure 5 is discussed, Figure 6 is indicated in the text) and in lines 241-254 (Figure 6 is discussed, Figure 7 is indicated in the text).

Author Response

Dear reviewer,

On behalf of all the contributing authors, I would like to express our sincere appreciations of your professional review work on our article (Manuscript ID: coatings-2545688). These comments are all valuable and helpful for improving our article. All the authors have seriously discussed about all these comments. According to your comments, we have tried best to modify our manuscript to meet with the requirements of the journal. In this revised version, changes to our manuscript within the document were all highlighted by using red colored text. Point-by-point responses to the comments are listed below this letter.

Reviewer 2 Report

In this manuscript, the authors studied the polycrystalline p-type β-Ga2O3 films grown through thermal oxidation of GaN at 1000 to 1100 3 °C in N2O atmosphere. The normalized X-ray diffraction (XRD), the high-resolution transmission electron microscopy (HRTEM), and the selected area electron diffraction (SAED) patterns were used to confirm the characteristic of polycrystalline and anisotropic growth. The manuscript should be accepted after addressing the following issues;

1)      In page 3, Figure 1a depicts the 104 PL spectra (at a wavelength ranging from 242 to 300 nm) of the samples obtained through 105 thermal oxidation at 1000, 1050, and 1100 °C……………….. the authors add more details related to the chage in intensity with the annealing temperatures.

2)      In XRD data, the authors should add more details about the XRD, such as crystalline size, lattice parameters etc.

3)      The authors should add Rietveld refinement of X-ray diffraction and the JCPDS card number.

4)      The authors also calculate the d-spacing from the XRD and compare it.

5)      On page 7, the authors mentioned, “The test mode of SIMS in this study was based on the relative content ratio elements…………. How the authors confirm the ratio

6)      In Figure 7, the mobility and carrier concentration is measured using the Hall measurement. The authors should add the information or criteria to calculate these parameters.

7)      For mobility and carrier concentration, the samples are measured in a vacuum. Did the authors check the other environment?

8)      What’s about the stability of the film? Did the author check the stability of the device?

9)      Many spelling and formatting typos in this paper and the authors should check and revise them thoroughly.        

Many spelling and formatting typos in this paper and the authors should check and revise them thoroughly.    

Author Response

(The authors gave the same response as above.)

Reviewer 3 Report

Referee report on manuscript “Further characterization of the polycrystalline p-type β-Ga2O3 films grown through thermal oxidation of GaN at 1000 to 1100 °C in N2O atmosphere

The article definitely contains some new and interesting results that can be recommended for publication, but only after the article has been finalized and some shortcomings have been eliminated.

1. Line 36. Cited reference [19] is absent in the list of references.

2.  The role of computer modeling, especially ab-initio calculations  in understanding and advancing various applications of Ga2O3 cannot be overestimated. So the fact that this material is important also from a theoretical point of view should be reflected.

Usseinov, A.; Koishybayeva, Z.; Platonenko, A.; Pankratov, V.; Suchikova, Y.; Akilbekov, A.; Zdorovets, M.; Purans, J.; Popov, A.I. Vacancy Defects in Ga2O3: First-Principles Calculations of Electronic Structure. Materials 202114, 7384. https://doi.org/10.3390/ma14237384

Zhang, C.; Li, Z.; Wang, W. Critical Thermodynamic Conditions for the Formation of p-Type β-Ga2O3 with Cu Doping. Materials 202114, 5161. https://doi.org/10.3390/ma14185161

Mondal, A.K.; Mohamed, M.A.; Ping, L.K.; Mohamad Taib, M.F.; Samat, M.H.; Mohammad Haniff, M.A.S.; Bahru, R. First-Principles Studies for Electronic Structure and Optical Properties of p-Type Calcium Doped α-Ga2O3Materials 202114, 604. https://doi.org/10.3390/ma14030604

3. Fig.1. The authors are sure that this is not a mistake. There is a feeling that this spectrum is an absorption spectrum.  Furthermore, no one has observed luminescence at 246 nm in gallium oxide!!! Moreover, the excitation wavelength is not indicated here and the excitation spectrum is not given.

4. Equation (1)  requires a description of the process model. In this form, it is not obvious. What does “Ectivation” mean ?

5. Lines 196-206. How can you prove the formation of oxygen and gallium vacancies? What charge state are they in?

6. in many cases, the authors give values with excessive precision, but this requires a special explanation. See, line 231, Table 1,  lines 273 and 303.

The work may be recommended for publication after adequate consideration of the above comments/questions.

Author Response

Dear reviewer,

Thanks for providing us with this great opportunity to submit a revised version of our manuscript (Manuscript ID: coatings-2545688). According to the your comments, we have carefully modified our manuscript to meet with the requirements of the journal. In this revised version, changes to our manuscript within the document were all highlighted by using red colored text. Point-by-point responses to the comments are listed below this letter.

Round 2

Reviewer 1 Report

I generally agree with the authors' answers to my comments, but there are still some questions:

1. From the author's answers and the concentration profiles of the elements presented in Fig. 1-3 in Appendix, I did not understand by what principle the authors determined the thickness of the Ga2O3 film. The proposed values do not correspond to any specific situation, for example, the end of the plateau on the oxygen profile, or the beginning of the plateau on the gallium or nitrogen profile.

2. In Fig. 4, the authors presented ln(Thickness) on the ordinate axis. It is clear that the logarithm is a dimensionless quantity, but its value in this case depends on the units in which "Thickness" is measured (probably, nanometers). You need to specify this either directly on the axis, or in the figure caption.

Author Response

Dear reviewer,

Thanks again for providing us with this great opportunity to submit a revised version of our manuscript (Manuscript ID: coatings-2545688). According to the your comments, in this revised version, the second round of changes to our manuscript within the document were all highlighted by using red colored text with the green background. Point-by-point responses to the comments are listed below this letter.

There are totally two content changes in the second round of revisions:

  • Line 238, in previous version: Based on these measurements,

Line 245, in current second round of revisions: Based on the FIB-TEM measurements [37],

  • Line 193, in current second round of revisions: in figure caption of Figure 4, the unit nm is added after the thickness.

In addition to the above two content amendments, according to the opinion of another reviewer, we submitted to MDPI for English polishing again.

Reviewer 2 Report

Accept in the present form

Author Response

Dear Editor,

Thank you very much for your positive comments and the acceptance decision on the revised version of the manuscript!

Reviewer 3 Report

After successful revision, this manuscript can be recommended for publication.

Author Response

Dear reviewer,

Thanks again for providing us with this great opportunity to submit a revised version of our manuscript (Manuscript ID: coatings-2545688). According to your comments, we have submitted the revised manuscript to MDPI (https://www.mdpi.com/authors/english) and completed another round of English polishing. Hope the quality of the revised manuscript after English polishing can meet the requirements.

I would like to express our sincere appreciations of your professional review work on our article.